# Influence of Eco-Friendly Processing Aids on Silica-Based Rubber Composites

## Sung Ho Song

Division of Advanced Materials Engineering, Kongju National University, Chungnam 330-717, Korea; shsong805@kongju.ac.kr

**Abstract:** As eco-friendly "green tires" are being developed in the tire industry, conventionally used carbon black is being replaced with silica in rubber compounds. Generally, as a lubricant and dispersing agent, processing aids containing zinc ions have been employed as additives. However, as zinc is a heavy metal, alternative eco-friendly processing aids are required to satisfy worldwide environmental concerns. Furthermore, non-toxic, degradable, and renewable processing aids are required to improve the mechanical properties of the rubber composites. In this study, we evaluated the effects of diverse silica-based processing aids containing hydrocarbon, benzene, and hydroxyl functional groups on the mechanical properties of rubber composites. Among them, rubber composites that used amphiphilic terpene phenol resin (TPR) with hydrophilic silica showed compatibility with the hydrophobic rubber matrix and were revealed to improve the mechanical and fatigue properties. Furthermore, owing to the enhanced dispersion of silica in the rubber matrix, the TPR/styrene butadiene rubber composites exhibited enhanced wet grip and rolling resistance. These results indicated that TPR had multifunctional effects at low levels and has the potential for use as a processing aid in silica-based rubber composites in tire engineering applications.

**Keywords:** terpene phenol resin; processing aids; hydrocarbon resin; alpha methyl styrene resin; alkyl phenol resin; rubber composites

---

## 1. Introduction

Rubber can be employed in many applications that require viscoelastic polymers and in a wide range of industrial fields by adding fillers, especially in the tire industry [1]. Styrene butadiene rubber (SBR), which is a commercial rubber that has been used as a general tread compound in the tire industry, is a nonpolar and noncrystalline copolymer with a low tensile strength. Therefore, various reinforcing fillers are required in the matrix of SBR composites to improve their mechanical, thermal, and electrical properties [2–5]. In general, carbon black (CB) is an important filler that is used for improving the mechanical properties of almost any industrial rubber, such as abrasion resistance, hardness, and tear strength. However, rubber composites using CB are limited by the lack of wet traction and rolling resistance. Furthermore, because CB is extracted from crude oil, its manufacturing process generates significant greenhouse gas emissions and waste. With the recent implementation of the European Tire Labelling Regulation [6], silica has been used as part of a "green tires" concept (a new breed of environmentally friendly tires with silica) in tread formulation, resulting in high wet grip and low rolling resistance [7,8]. Furthermore, silica-filled rubber composites exhibited higher stiffness, modulus, tensile strength, lower heat build-up, and better resistance to wear and tear of the tire compounds [8–10]. However, silica, with silanol and siloxane functional groups, differs from CB in its chemical composition and conventional method of manufacturing the composites [11]. The silanol group is an acid [12] that interacts with basic accelerators to slow the curing rate and lower the cross-link density in sulfur curing systems [13]. The viscosity of the composite increases with the amount of silica, which tends to

---

interact strongly on their own due to their polar and hydrophilic properties [14]. Hence, manufacturing composites with silica is more difficult due to poor dispersibility caused by strong polar bonds between the filler particles when made from non-polar rubbers [15–18]. Silica coupling agents, such as bis(triethoxysilylpropyl)disulfide (TESPD) and bis(triethoxysilypropyl)tetrasulfide (TESPT), have been used to enhance filler–rubber interactions by modifying the silica surface. These act as chemical bonding agents between the rubber and silica in the vulcanization process [19–21]. Furthermore, processing agents are used in the manufacture of tire rubber composites. The processing aids include hydrocarbons, fatty acid derivatives, low-molecular-weight polymers, synthetic resins, and other organic compounds, which are used as lubricants and dispersion agents, according to their chemical structure [22]. Bi-functional silanes are used in the mixing process to improve the processability and the chemical bonding between the rubber and silica. In addition, various zinc soaps, including zinc naphthenate, zinc stearate, and zinc resinate, are used as intermolecular lubricants, owing to their ability to enhance the rubber solubility in rubber composites. However, because zinc consumption is an environmental problem, the major contributors are under pressure to reduce its use and develop a substitute. Thus, it is necessary to find environmentally friendly, alternative processing aids for use in rubber composites. Recently, rice bran oil and rice oil have been used for polychloroprene (CR) [22], acrylonitrile butadiene (NBR), and SBR [23] as processing aids. Furthermore, castor oil has been used in nitrocellulose, polystyrene rubber, and natural rubber [24,25]. Moreover, soybean oil was used as a plasticizer in NR and ester gums [24]. Kukreja et al. found that adding palm oil in an NBR matrix improved the aging resistance and plasticizing efficiency [26]. Asharf et al. reported that poly(methyl methacrylate) (PMMA) composites with palm oil exhibited enhanced mechanical properties [27]. However, because processing aids, such as natural oils, have a higher molecular weight, the hydrophobic characteristic of the oil greatly hinders its dispersion and reactions with the silica and rubber matrices in the tread compound. Furthermore, it has been less studied for zinc-free processing aids in terms of the improved mechanical properties, such as modulus, tensile strength, fatigue, and abrasion properties, using a novel processing aid.

In this study, we investigated the effects of new silica-based processing aids, utilizing terpene phenol resin (TPR), which has hydrophilic and hydrophobic groups at opposite ends of the molecule. Whereas the hydrophobic group reacts with the double bonds in the rubber chain [28,29], the hydrophilic group reacts with silanols on the silica surface [30,31]. A master batch made of diverse processing aids and silica fillers was used to improve the dispersion in the rubber latex matrix through latex mixing methods. This processing method was a fast, convenient, versatile, inexpensive, and scalable method for improving the performance of rubber composites. The rubber composites with TPR had better mechanical, fatigue, and abrasion properties than composites with other processing aids. Furthermore, the TPR–rubber composite exhibited enhanced wet and dry braking and rolling resistance due to the improved dispersion of silica in the rubber matrix.

## 2. Materials and Methods

### 2.1. Materials

An SBR (SBR 1500, Kumho Petrochem Co. Ltd., Seoul, Korea) was used as the matrix material, which consisted of 23% styrene and 77% butadiene. The silica used in this study was obtained from Rhodia Silicas Co., Incheon, Korea, and the carbon black (N-330) was supplied by OCI Co. Ltd., Seoul, Korea. N-tert-butyl-benzothiazole sulfonamide (TBBS) was obtained by Shandong Shanxian Co. Ltd., Shandong, China. Zinc oxide (ZnO), stearic acid (S/A), sulfur, and oleic acid were purchased from Sigma–Aldrich, Seoul, Korea. Hydrocarbon resin (HCR), alkyl phenol resin (APR), alpha methyl styrene resin (AMR), and TPR was obtained from NEXEN Tire Co. Ltd., Yangsan, Korea.

## 2.2. Preparation of the Rubber Composites

The SBR composites were fabricated by following the standard procedures in Table 1. First, an SBR latex of 40 parts per hundred parts rubber by weight (phr) and 30 phr silica, as well as diverse processing aids, were mixed using vigorous stirring for 24 h. Then, the SBR emulsions were coagulated with a sulfuric acid solution. The coagulated composites were rinsed with water until a pH of 6–7 was achieved, and then the composites were dried in an oven at 50 °C for 24 h. The master batches were mixed from the SBR (60 phr) and silica (30 phr)/SBR (40 phr) emulsion, as well as CB (20 phr) in a Banbury mixer at a rotor speed of 60 rpm. Finally, the vulcanization agents and additives were added at the end of the mixing to initiate the curing process. The resulting materials were placed in an aluminum mold and cured at 160 °C to reach $T_{90}$ under pressure, as measured using a rheometer.

**Table 1.** Formulation of rubber composites with varying content fractions of different resins in parts per hundred parts of rubber (phr).

| STEP | Materials | HCR 1.5/3.0/6.0 | AMR 1.5/3.0/6.0 | APR 1.5/3.0/6.0 | TPR 1.5/3.0/6.0 | Injection Materials | Injection Time |
|------|-----------|-----------------|-----------------|-----------------|-----------------|---------------------|----------------|
| Step 1 | SBR Latex | 50 | 50 | 50 | 50 | | |
| | Silica | 40 | 40 | 40 | 40 | | |
| | HCR | 1.5/3.0/6.0 | | | | SBR/SBR Latex | 0 |
| | AMR | | 1.5/3.0/6.0 | | | Carbon Black | 30 |
| | APR | | | 1.5/3.0/6.0 | | Master Batch | 100 |
| | TPR | | | | 1.5/3.0/6.0 | Chemical | 180 |
| Step 2 | SBR | 50 | 50 | 50 | 50 | Drop | 300 |
| | Carbon Black | 20 | 20 | 20 | 20 | | |
| | S/A | 1 | 1 | 1 | 1 | | |
| | Sulfur | 1.75 | 1.75 | 1.75 | 1.75 | | |
| | ZnO | 2 | 2 | 2 | 2 | | |
| | TBBS | 1 | 1 | 1 | 1 | | |

## 2.3. Characterization

The curing characteristics were measured using a moving die rheometer (RPA, TA instruments Co., New Castle, DE, USA). The tensile strength and elongation tests were performed on an Instron tensile machine (Auto 750, Instron Co., Norwood, MA, USA) with a crosshead speed of 300 mm/min. Dumbbell-shaped test specimens (fixed rubber dumbbell type) with a prepared 100 mm thickness and 5 mm width underwent at least four tests for each case. The fatigue properties of rubber composites were investigated using a DeMattia Flex Tester (Worldoftest Co., Ontario, ON, CA) and dynamic tests were performed in tensile mode using a dynamic mechanical analyzer (model DMA 50N01Db, Metravib, Limonest, France). The loss factor and the storage modulus (tan δ, defined as the ratio of the loss modulus to the storage modulus) as a function of the temperature (temperature sweep) were measured from 20 °C to 120 °C at a heating rate of 3 °C/min at a frequency of 10 Hz. Transmission electron microscopy (TEM, JEOL JSM-2100, Tokyo, Japan) was used to observe the morphology of the fractured surface of rubber composites to confirm the dispersibility of silica. Heat capacity was determined through a differential scanning calorimetry (DSC, D8000, PerkinElmer, Songdo-dong, Korea)

## 2.4. Measurement of Swelling

The swelling tests were conducted by immersing a specific weight of the rubber composite with various processing aids in toluene at room temperature for 24 h and the equilibrium swelling of toluene (Q%) was determined using the standard method (ASTM D471-06). Each sample prepared from the slabs used to cut specimens for mechanical properties testing was weighed and immersed in excess toluene at 30 °C. The toluene absorption rate was measured by drying the surface of the sample after a set period and weighing it, where the evaporation of the solvents was negligible and the scattering of the measurements was less than 1%.

## 3. Results and Discussion

Rubber composites of silica and SBR matrix were prepared with a variety of processing aids via a latex mixing method. Most processing aids contain benzene or other ring structures, such as phenolic resin, a nonpolar hydrocarbon, and polar hydroxyl functional groups from Figure 1. Thus, while HCR and AMR with a hydrocarbon and a benzene ring have hydrophobic properties, the APR and TPR with a hydrocarbon, a benzene ring, and hydroxyl (–OH) groups have amphiphilic properties and this structure serves to enhance the dispersion of silica in the rubber matrix. Most resins with a benzene ring are more restricted to being a rubber chain than other resins with hydrocarbon groups [32].

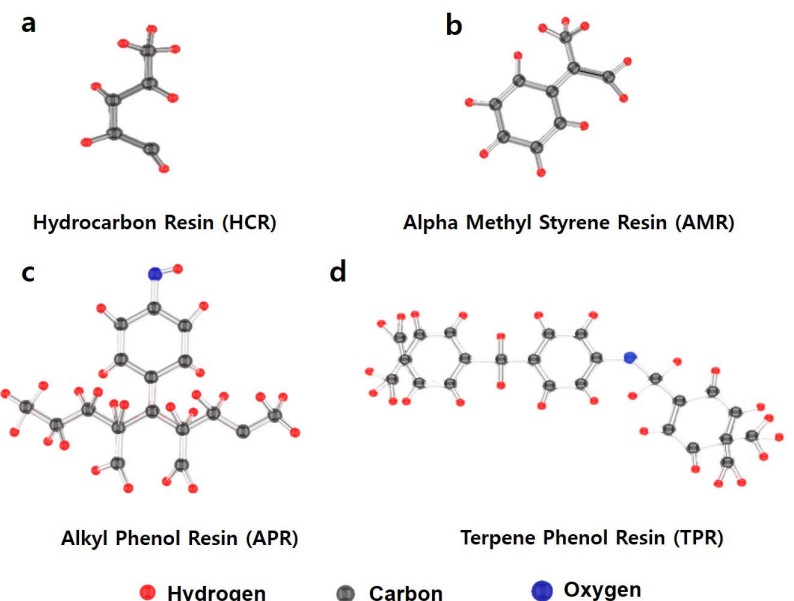

**Figure 1.** Molecular structure diagram of (**a**) hydrocarbon resin, (**b**) alpha methyl styrene resin, (**c**) alkyl phenol resin, and (**d**) terpene phenol resin.

The processability, which is important during the formulation, was determined by evaluating the curing properties, such as the cure time, minimum torque, and maximum torque. Figure 2 shows the curing properties, based on the cure time ($t_{40}$ and $t_{90}$), of rubber composites filled with various processing aids. The times required for 40% and 90% cure completion were designated as the $t_{40}$ and $t_{90}$ times, respectively. The amount of added processing aids was small; consequently, the thermal conductivity did not change and there was no distinct difference in the curing time of rubber composites with various processing aids, as shown in Figure 2a,b. However, as the content of the processing aids with very low thermal conductivity increased, $t_{40}$ and $t_{90}$ tended to increase. Additionally, a rheometer is commonly used as an indicator of the processing safety of rubber composites. The SBR composite with TPR exhibited a reduced Mooney viscosity, which facilitated the preparation process of the rubber composites, as shown in Figure 2c.

Figure 3 shows a comparison of the mechanical properties of the SBR composites with different added processing aids. A comparison of the torque in the SBR composites was investigated as a function of the increasing, diverse processing aids. The $M_H$ (torque maximum) − $M_L$ (torque minimum) (Δ torque) value, which is the cross-link density of vulcanization, of the SBR composite decreased as the processing aid contents increased (Figure 3a–c); furthermore, the Δ torque value of the SBR composite with TPR (27.3 N·m) at 1.5 phr was higher than that of other composites. This result means that the silica dispersion and stronger cross-linking with the SBR during vulcanization were enhanced by the TPR with hydrophilic and hydrophobic groups because the interaction between hydrophilic silica and hydrophobic rubber increased. Figure 3d,e shows the modulus and tensile strength of the SBR composites with different processing aids as the processing aid contents increase. The modulus

and tensile strength (T/S) of the composites with TPR and AMR at 3.0 phr increased by as much as 36% and 3%, respectively. This improvement was attributed to the enhanced dispersion and strong interfacial bonding of silica in the rubber matrix by TPR with amphiphilic functional groups. Moreover, the modulus and tensile strength of the TPR/SBR composite were higher than those of the APR/SBR composites due to the increased interaction between the benzene functional groups of TPR and SBR. The elongation at break of the SBR composite decreased with increasing amounts of processing aids, while the elongation increased with increasing viscosity. The elongation at break of the TPR/SBR composite (473%) was lower than that of the HCR/SBR composite (487%) due to stronger interfacial bonding and the restriction of the segmental elastomer chain motion [33].

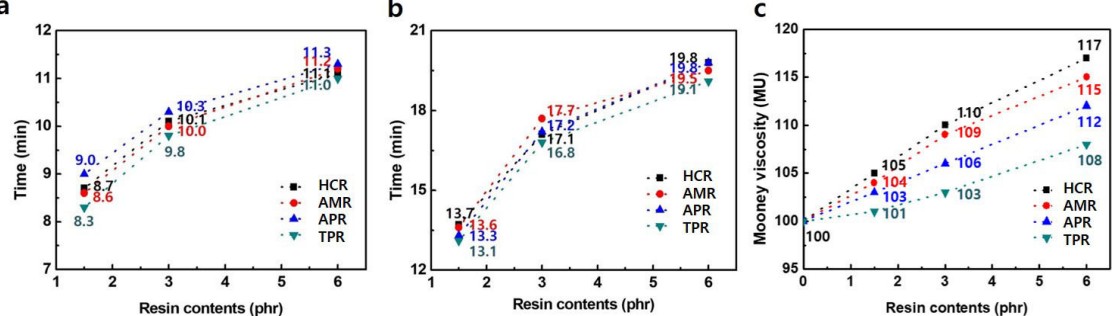

**Figure 2.** Curing properties of rubber composites: (**a**) curing properties ($t_{40}$) of the rubber composites with varying content fractions of different resins, (**b**) curing properties ($t_{90}$) of the rubber composites with varying content fractions of different resins, and (**c**) Mooney viscosity of the rubber composites with varying content fractions of different resins.

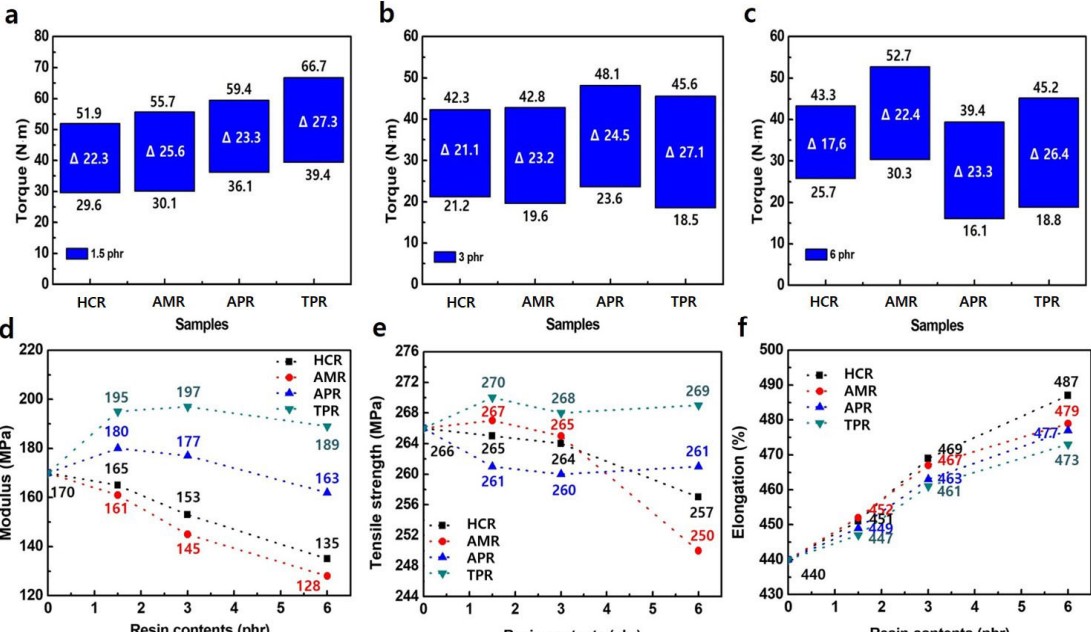

**Figure 3.** Mechanical properties of the rubber composites: (**a**) torque of the rubber composites with different resins at 1.5 phr, (**b**) torque of the rubber composites with different resins at 3.0 phr, (**c**) torque of the rubber composites with different resins at 6.0 phr, (**d**) elastic modulus of the rubber composites with varying content fractions of different resins, (**e**) tensile strength of the rubber composites with varying content fractions of different resins, and (**f**) elongation of the rubber composites with varying content fractions of different resins.

To further investigate the mechanical properties, swelling tests of rubber composites with different processing aids were carried out, with the results shown in Table 2. The equilibrium swelling Q% was calculated as follows:

$$Q = [(Ws - Wd)/Wd] \times 100\%, \tag{1}$$

where Ws is the weight of the swelled specimen, and Wd is the weight of the dried specimen. The cross-linking densities (V, mol/cm$^3$) of the SBR composites were found via the solvent swelling measurements using the Flory–Rehner equation [34,35], as follows:

$$V = 1/2 \times Mc, \tag{2}$$

where Mc is the molecular weight between the cross-links (g/mol) and:

$$Mc = -\varrho \times Vs(Vr^{1/3} - 2Vr/f)/[\ln(1 - Vr) + Vr + \chi \times Vr^2], \tag{3}$$

where Vr is the volume fraction of the swollen rubber, Vs is the molar volume of the toluene, $\chi$ is the interaction parameter of rubber, f is the network functionality, and $\varrho$ is the density of rubber.

**Table 2.** Swelling and cross-linking density characteristics of rubber composites with varying content fractions of different resins.

| Materials | HCR 3.0 phr | AMR 3.0 phr | APR 3.0 phr | TPR 3.0 phr |
|---|---|---|---|---|
| Equilibrium swelling (%) | 145.13 ± 0.3 | 152.12 ± 0.2 | 165.72 ± 0.4 | 173.14 ± 0.3 |
| Crosslinking density ($10^4$ mol/g) | 2.35 ± 0.4 | 2.75 ± 0.5 | 2.87 ± 0.3 | 2.98 ± 0.5 |

These results indicate that the functional groups of TPR could provide the chemical cross-link and physical entanglements for the cross-linking network between the silica and rubber matrix. Generally, the rubber chains were immobilized on the filler surface as a constrained area between the rubber matrix and the filler due to a processing agent, thereby preventing the enhanced reinforcement of fillers and phase separation in rubber composites [36,37]. The weight fraction of rubber chains fixed on the filler surface ($\chi$im) was determined as follows [38]:

$$\Delta Cpn = \Delta Cp/(1 - \omega), \tag{4}$$

$$\chi im = (\Delta Cp0 - \Delta Cpn)/\Delta Cp0, \tag{5}$$

where $\Delta Cp$ is the heat capacity jump at the glass transition temperature (Tg), $\Delta Cp0$ is the heat capacity jump at the Tg of the neat rubber, $\Delta Cpn$ is the heat capacity jump normalized to the rubber fraction, and $\omega$ is the weight fraction of fillers in the rubber composites.

These parameters were determined through a differential scanning calorimetry (DSC) experiment. The value of $\chi$im of the TPR/SBR composite was higher than that of other rubber composites, as shown in Table 3. The higher the value of $\chi$im, the stronger the interfacial interaction between the filler and the rubber matrix. The improved reinforcement could be attributed to enhanced interfacial adhesion between the silica and the rubber, and the restriction of rubber segment chain movement was due to the improved dispersion of silica in the rubber with added TPR.

**Table 3.** Parameter values of rubber composites with varying content fractions of different resins.

| Materials | $\omega$ | $\Delta Cp$ | $\Delta Cpn$ | $\chi im$ |
|---|---|---|---|---|
| HCR 3.0 phr | 3.0 | 0.485 | 0.5 | 20.0 |
| AMR 3.0 phr | 3.0 | 0.451 | 0.465 | 25.6 |
| APR 3.0 phr | 3.0 | 0.467 | 0.481 | 23.0 |
| TPR 3.0 phr | 3.0 | 0.423 | 0.436 | 30.2 |

Furthermore, transmission electron microscopy (TEM) provided direct evidence for evaluating the dispersion of fillers and the phase morphology in relation to the mechanical properties of rubber composites. The TEM images of the fractured rubber composites are exhibited in Figure S1. In the rubber composite with HCR, these silica fillers were observed as globular aggregates of various sizes. However, for the rubber composites with TPR, large aggregates were not observed anymore but the silica was well distributed throughout the rubber matrix in Figure S1.

The dynamic mechanical properties of engineering rubber are generally important because many forms of rubbers are subjected to dynamic loads. One such property, the Tg, is shown as a function of resin contents in Figure 4a. With the addition of TPR in the SBR composite, the Tg was higher than that of HCR, i.e., the Tg was −29.6 °C and −35.8 °C at 3 phr, respectively, as shown in Figure 4a. Furthermore, the Tg of the TPR/SBR composite was higher than that of the APR/SBR composite. As the content of processing aids increased, Tg tended to decrease slightly. The dynamic mechanical properties (DMA) of SBR composites with different processing aids versus temperature are shown in Figure 4b,c. The DMA test to investigate tan δ (the ratio of loss modulus to storage modulus) showed the following physical properties of SBR composites, which was used to predict both the wet grip and rolling resistance properties [39,40]. The 0 °C tan δ of TPR (3 phr)/SBR composites, which indicates the pneumatic grip tire performance, was about 3.5% higher than that of HCR in Figure 4b. The 60 °C tan δ of SBR composites with TPR (3 phr), which indicates the pneumatic rolling resistance, was approximately 92% lower than that of the SBR composite with HCR in Figure 4c. Thus, the TPR/SBR composite had immense potential for improving wet grip and rolling resistance, and these results were consistent with the mechanical properties in Figure 2.

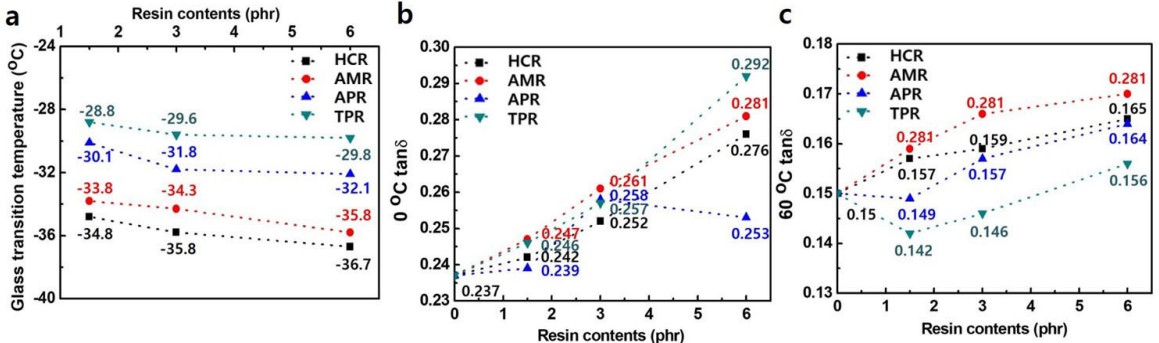

**Figure 4.** Dynamic properties of the rubber composites: (**a**) glass transition temperature (Tg) of the rubber composites with varying content fractions of different resins, (**b**) 0 °C tan δ of the rubber composites with varying content fractions of different resins, and (**c**) 60 °C tan δ of the rubber composites with varying content fractions of different resins.

The abrasion properties of an SBR composite with different processing aids with increasing contents are shown in Figure 5a–c. The abrasion properties of the rubber composites tended to decrease with the content of processing aids due to an increase in the viscosity and a decrease in mechanical properties. A large amount of HCR/SBR composite debris was found among all samples on the worn surfaces, corresponding to poor wear resistance, as shown in Figure 5a–c. Moreover, the abrasion properties of the TPR/SBR composites were much better than those of the HCR/SBR composites. The higher abrasion properties were attributed to the functional groups of the TPR and the siloxanes, as well as silanols on the silica, which provided chemical cross-links and physical entanglements within the cross-linked network [41]. The fatigue properties of the SBR composites (Figure 4c and Table 4) were measured after 2000, 6000, and 10,000 cycles. The fatigue crack growth (dc/dn) was calculated as follows [42]:

$$dc/dn = A \times G^{\alpha} \qquad (6)$$

where n is the number of loading cycles, c is the crack length, and G is the tear energy.

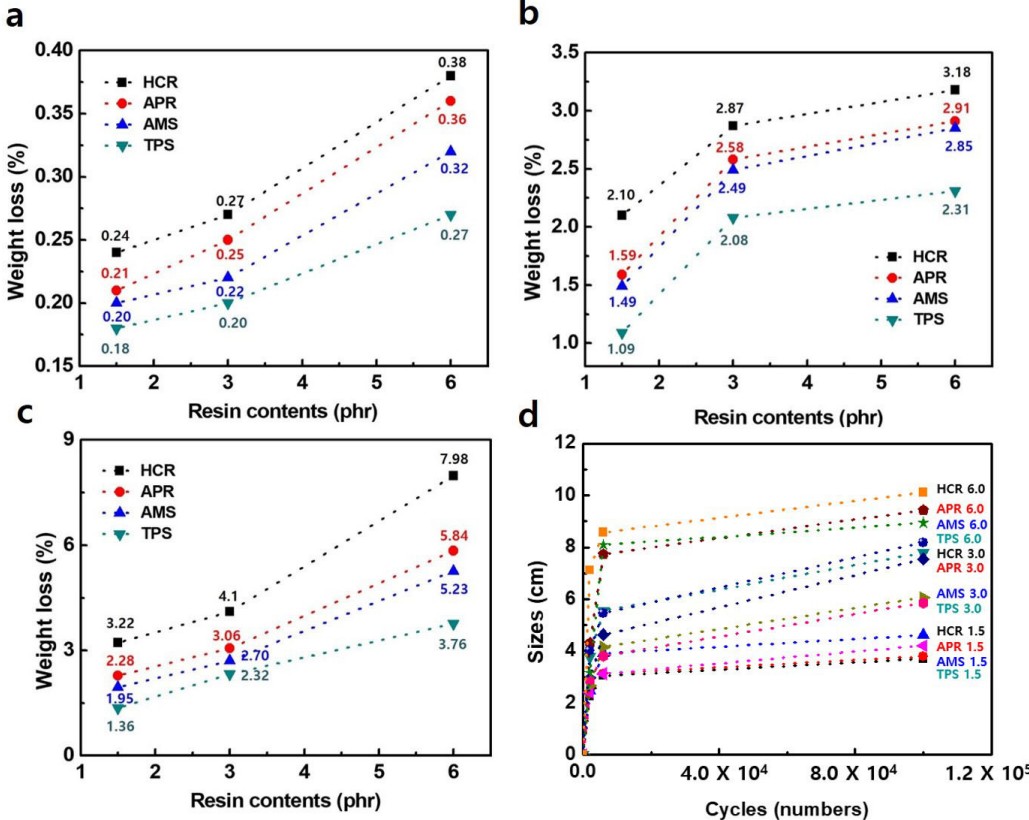

**Figure 5.** Mechanical properties of rubber composites: (**a**) Akron abrasion of the rubber composites with varying content fractions of different resins, (**b**) Lamborn abrasion of the rubber composites with varying content fractions of different resins, (**c**) William abrasion of the rubber composites with varying content fractions of different resins, and (**d**) Fatigue properties of the rubber composites with varying content fractions of different resins.

**Table 4.** Fatigue properties of rubber composites with varying content fractions of different resins.

|       | HCR 1.5 | HCR 3.0 | HCR 6.0 | APR 1.5 | APR 3.0 | APR 6.0 | AMS 1.5 | AMS 3.0 | AMS 6.0 | TPR 1.5 | TPR 3.0 | TPR 6.0 |
|-------|------|------|-------|------|------|------|------|------|------|------|------|------|
| 0     | 0    | 0    | 0     | 0    | 0    | 0    | 0    | 0    | 0    | 0    | 0    | 0    |
| 2000  | 2.47 | 3.78 | 7.12  | 2.42 | 2.95 | 4.31 | 2.36 | 2.67 | 3.16 | 2.26 | 2.87 | 4.01 |
| 6000  | 3.92 | 5.56 | 8.57  | 3.13 | 4.63 | 7.73 | 3.13 | 4.16 | 8.09 | 3.05 | 3.81 | 5.46 |
| 10,000 | 4.61 | 7.79 | 10.11 | 4.21 | 7.54 | 9.43 | 3.79 | 6.07 | 8.94 | 3.7  | 5.86 | 8.19 |
| dc/dn | 3.24 | 5.11 | 9.07  | 3.21 | 5.07 | 8.12 | 3.17 | 4.11 | 7.35 | 3.12 | 4.02 | 6.25 |

Generally, fatigue crack growth is proportional to the required tear energy, and the higher the required tear energy, the better the fatigue properties. The fatigue crack growth decreased as the processing aid contents increased, as shown in Figure 5d. Additionally, the fatigue crack growth of the SBR composite (5.11) with HCR at 3 phr was over 27% lower than that of the SBR (4.02) composite with TPR, as shown in Figure 5d and Table 4. These features make the composites very promising for green tire applications.

## 4. Conclusions

In this study, the effects of various silica-based processing aids, including hydrocarbon, benzene, and hydroxyl functional groups, were evaluated for the development of environmentally friendly "green tires" using silica-based compounds. Importantly, through the incorporation of TPR into an SBR matrix containing a silica filler, the silica was uniformly dispersed in the rubber matrix and the interfacial bonding between the silica and SBR was strengthened, which remarkably enhanced

the mechanical and fatigue properties. Furthermore, the amphiphilic TPR with hydrophobic and hydrophilic functional groups was incorporated into the rubber matrix using a new, environmentally friendly, versatile, and simple approach based on latex technology. Notably, the TPR/SBR composites exhibited significantly enhanced abrasion, wet braking, and rolling resistance due to improved dispersion of the silica in the rubber matrix, which demonstrated the effectiveness of the low loadings of TPR in the rubber composites. The rubber composites prepared with TPR offer new opportunities for applications in tire engineering as materials for "green tires".

**Supplementary Materials:** The following are available online at http://www.mdpi.com/2076-3417/10/20/7244/s1, Figure S1: (**a**) TEM images of rubber composites with HCR. (**b**) TEM images of rubber composites with TPR.

**Funding:** This research was supported by the Basic Science Research Program through the National Research Foundation of Korea (NRF), which was funded by the Ministry of Education (NRF-2019R1A6A1A03032988, NRF-2017R1C1B507647614, and NRF-2020R1I1A307162811).

**Acknowledgments:** We thanks NEXEN Tire for fabricating the rubber nanocomposites.

**Conflicts of Interest:** The author declares no conflict of interest.

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
