# Peer review of "Influence of Eco-Friendly Processing Aids on Silica-Based Rubber Composites"

_applsci, doi:10.3390/app10207244_

Round 1

Author Response

Dear Editor,

We thank the reviewers for their comments and kind suggestions. We are also glad to send this clarification report to the editor and reviewers. Our manuscript has been explicitly revised in accordance with the reviewers’ comments and suggestions, and the modifications are marked with red color in the revised manuscript.

We believe that these modifications make our manuscript more appropriate for publication in Applied Sciences. We also anticipate that the novel proposed approach and results presented in this manuscript will be of significant interest to the general readership of the journal.

The response letter for addressing the reviewers’ comments consists of 10 pages.

Thank you and best regards,

Yours sincerely

Prof. Sung Ho Song

Reviewer 1

Comments: This manuscript describes new processing aids for silica-based rubber composites. The contents and the experimental results are very attractive and interesting. However, it should be mentioned that quality as a scientific article has to be improved, including English expressions. I concluded this paper is possibly publishable, but major revision must be needed.

  1. 1. I found some errors in English expressions. The authors need to be checked by a native English speaker.

Response for question 1

Author is sorry for the language problem as mentioned by Reviewer. To improve English in the manuscript, the manuscript was carefully reviewed again and had a proofreading from a native speaker and professional editing service in KAIST. All of the revised parts were remarked in the main text.

  1. Some tables are not correctly printed. I cannot read out data.

Response for question 2

Thank you for your comments. Following the reviewer’s suggestion, I checked the table in the manuscript again, but the contents related to the table were well mentioned in the manuscript. Also, according to the MDPI format, the table cannot be attached as a picture file, making it difficult to modify. Therefore, the form of a picture file to help reviewers understand.

Table 1. Formation of rubber composites with varying content fraction of different resins.

                                                                                             phr

Table 2. Swelling and cross-linking density characteristics of rubber composites with varying content fraction of different resins.

Table 3. Parameter values of rubber composites with varying content fraction of different resins.

Table 4. Fatigue properties of rubber composites with varying content fraction of different resins.

  1. References have severe problems as follows:

2-5: These should be replaced with a book such as “Rubber Compounding chemistry and applications” ed. By B. Rodgers, CRC Press.

Response for question

Thank you for your comments. However, in reference 2-4, paper in which various fillers were applied to the recently studied rubber was attached to the reference. Although a book is good, it seems that it may be cited as references.

Introduction: Therefore, various reinforcing fillers are required in the matrix of SBR composites to improve their mechanical, thermal an electrical properties [2–5].

Reference 2: About the rubber and carbon nanofiber decorated with iron oxide

Reference 3: About the rubber and carbon nanotubes

Reference 4: About the rubber and clay nanplatelets

6: This is not appropriate. It should be some official articles or (web) documents by the European Commission.

  1. D’Apuzzo, M.; Evangelisti, A.; Nicolosi, V. An exploratory step for a general unified approach to labelling of road surface and tyre wet friction. Accid. Anal. Prev. 2020, 138, 105462.

7, 8: Are they suitable?

In roger to improve the wet grip, a tire with silica is generally applied, but hydrophilic silica is less reactive with hydrophobic rubber. Therefore, recently, in order to make hydrophobic rubber hydrophilic, many studies are underway to make a composite by modifying the end of the rubber.

Therefore, they are suitable.

  1. The reviewer cannot obtain it. It should be a more general journal.
  2. Sattayanurak, S.; Noordermeer, J.W.M.; Sahakaro, K.; Kaewsakul, W. Dierkes, K.; Blume. A. Silica-Reinforced Natural Rubber: Synergistic Effects by Addition of Small Amounts of Secondary Fillers to Silica-Reinforced Natural Rubber Tire Tread Compounds. Adv. Mater. Sci. Eng. 2019, 2019, 1–8.

12, 18, 22, 24: These are suitable and necessary.

31: Is this correct?

  1. Huang, R.; Pan, Q.; Feng, K. Maleic anhydride modified dicyclopentadiene resin for improving wet skid resistance of silica filled SSBR/BR composites. Appl. Sci. 2020, 10, 4478–4490.
  1. Is this correct? I could not find the eq (4) and (5).

The constrained region (rubber chains immobilized on the surface of fillers) serves as a commercializer in rubber composites. This reinforcement is enhanced by the filler and prevents phase separation, and the weight fraction of rubber chains immobilized on the surface of fillers (χm) can be calculated. In this paper, ΔCp, ω, χ can be obtained through DSC analysis. Therefore, this paper is suitable and necessary. Furthermore, it is possible to easily derive the mentioned equation in the manuscript by modifying the equation given in this paper.

  1. Is this correct? I could not find the eq (6)
  1. Han, Q.; Zhang, L.; Wu, Y. Relationship between dynamic fatigue crack propagation properties and viscoelasticity of natural rubber/silicon rubber composites. RSC Adv. 2019, 9, 29813-29820.

  1. Other detail points:

Response for question 4

Thank you for your good comments. All the content mentioned by the reviewer was revised and reflected in the manuscript. All of the revised parts were remarked in the main text.

L35: manufacturing process generates

L37: as a “green tires” concept (a new breed of environmentally friendly tire with silica) in tread formulation

L40-41: silica, with silanol and siloxane functional groups, differs from CB in its chemical composition and conventional method of manufacturing the composites

L62-64: This is a commonly used expression when citing other papers when writing a paper.

L79: In experimental part: Scanning electron microscopy (SEM, JEOL JSM-6460LV) was used to observe the morphology of the fractured surface of rubber composites to confirm the dispersibility of silica.

Results and discussion part:

Furthermore, transmission electron microscopy (TEM) provides direct evidence for evaluating the dispersion of fillers and phase morphology in relation to the mechanical properties of rubber composites. The TEM images of the fracture rubber composites are exhibited in Fig. S1. In the rubber composite with HCR, these silica fillers are observed as globular aggregates of various sizes. However, with the rubber composites with TPR, the large aggregates are not observed any more but the silica occurs to be well distributed throughout the rubber matrix in Fig. S1.

Fig. S1.

Fig. S1. a) TEM images of rubber composites with HCR. b) TEM images of rubber composites with TPR.

L79-80: This study demonstrates the advantages and potential of silica-based fillers in tire engineering.

2.2. Preparation of rubber composites.

I could not understand this subsection at all, including Table 1. A major improvement is necessary. I dought the authors misuse the unit “phr.” The value for polymer should be 100 phr in each process. The authors’ expression is so complicated.

Response for question

As mentioned by the reviewer. The total rubber used is 100 phr including latex rubber. Also, latex was used and 2 steps were applied for silica dispersion in the manuscript. The repression about the process is not complicated and general.

L99: Curing characteristics were measured by a moving die rheometer (RPA, TA instruments Co., UK).

L103-104: Tensile strength and elongation tests were performed on an Instron tensile machine (AutoX750, Instron Co., UK)

L104: Dumbbell-shaped test specimens (fixed rubber dumbbell type) prepared 100 mm thickness and 5 mm width were performed at least four tests for each case.

L106: DeMattia (DeMattia Flex Tester, Worldoftest Co., USA)

L117-118: “scattering” means errors.

L120: Rubber composites

L132: scorch time (t40; lapsed time for the increase of Mooney units by 40)

L137-138: rheometer

L148: The MH (Torque maxium)-ML (Torque minimum) (Δ torque) value…

L150: Δ torque value of the SBR composite with TPR (27.3 N·m) at 1.5 phr

L151-152: because the interaction between hydrophilic silica and hydrophobic rubber increase.

L154-155: The modulus and tensile strength (T/S) of the composites with TPR and AMR at 3.0 phr increased by as much as 36% and 3%, respectively.

L155-156: because the interaction between hydrophilic silica and hydrophobic rubber increase.

L157-L159:

  1. Tian, C.; Feng, Y.; Chu, G.; Lu, Y.; Miao, C.; Ning, N.; Zhang, L.; Tian, M. Interfacial nanomechanical properties and chain segment dynamics of fibrillary silicate/elastomer nanocomposites. Compos. B. Eng. 2020, 193, 108048.

Figure 3’s caption: Mechanical properties of rubber composites a) Torque of rubber composites with different resins at 1.5 phr. b) Torque of rubber composites with different resins at 3.0 phr. c) Torque of rubber composites with different resins at 6.0 phr. d) Elastic modulus of rubber composites with varying content fraction of different resins. e) Tensile strength of rubber composites with varying content fraction of different resins. f) Elongation of rubber composites with varying content fraction of different resins.

Table 2: The table is separated into two pages.

I don’t think it is necessary. Table 2 is composed of 3 lines and 5 columns, so it can be sufficiently expressed on one page.

L180-181: “These results indicate that the functional groups of TPR can provide the chemical cross-link and physical entanglements for the cross-link network between the silica and rubber matrix.”: Can you say the same thing for other resin such as HCR, AMR, and APR?

We investigated the effects of new silica-based processing aids, utilizing terpene phenol resin (TPR), which have hydrophilic and hydrophobic groups at opposite ends of the molecule in SBR during mixing and after vulcanization. Whereas the hydrophobic group reacts with the double bonds on the rubber chain, the hydrophilic group reacts with silanols on the silica surface. However, because, HCR has only hydrophobic group, and AMR and APR contain a relatively small amounts of hydrophilic group, no improved properties can be expected compared TPR.

L187: “neat rubber” means “raw rubber”.

L192-194: We added the TEM results.

L205: 3.5%

L207: 92%

L209: In general, dynamic properties tend to increase as the crosslinking density increases. Also, as the crosslinking density increase, the modulus and tensile strength increase, and the results are confirmed in Table 2 and 3 [39-40].

L202-204: Generally, the cyclical deformations of the tire tread compounds due to rotation and braking can be approximated to a process with a constant power supply but using different temperatures and frequencies. It is well known that the hysteresis of tread composites, characterized by tan δ 60 ℃, is a key parameter that exhibits a good correlation with the tire rolling resistance. Also, the value of tan δ 0 ℃ is related to the high frequency nature of the dynamic strain involved in the wet grip.

  1. Sengloyluan, K.; Sahakaro, K.; Dierkes, W. K.; Noordermeer, J. W. M. Silica-reinforced tire tread compounds compatibilized by using epoxidized natural rubber. Eur. Polym. J. 2014, 51, 69-79.
  2. Li, Y.; Han, B.; Wen, S.; Lu, Y.; Yang, H.; Zhang, L.; Liu, L. Effect of the temperature on surface modification of silica and properties of modified silica filled rubber composites. Compos. Part A. Appl. Sci. Manuf. 2014, 62, 52-59.

L219-222: the abrasion properties of the TPR/SBR composites is much higher than that of HCR/SBR composites.

L224: where, n is the number of loading cycles, c is the crack length and G is the tear energy.

Eqs. (6)

Table 4 completely destroyed. Is the raw on the extreme left ”cycle” ?

Also, according to the MDPI format, the table cannot be attached as a picture file, making it difficult to modify. Therefore, the form of a picture file to help reviewer understand.

L231: The term 'green tyre' refers to a whole new breed of environmentally friendly rubber with silica. By the incorporation of TPR with amphiphilic properties into an SBR matrix containing silica filler, it is uniformly dispersed with silica in the rubber matrix and the interfacial bonding between the hydrophilic silica and hydrophobic SBR is strengthened, which remarkably enhances the mechanical and fatigue properties.

L252: The approach described herein has identified new opportunities for the development of new scalable, commercial, and high-performance rubber engineering products

L255: The authors are expected to write more detail.

The caption in Fig. 5. is composed of the overall experimental results and the caption for the detailed experimental content such as a) Akron abrasion, b) Lamborn abrasion and c) William abrasion.

Reviewer 2 Report

The manuscript present an influence of some chemical compounds in the obtaining of rubber composites to improve dynamic mechanical properties. Despite of interesting data, figures and discussion there are and some gaps from my point of view.

Abstract:10-13...zinc? inappropriate...that in manuscript is not so developed about it...

introduction: at least form 70 it is necessary to rewrite the text. You have already here and conclusions (77-80) 

2.3.  Characterization! mechanical, structural

2.4. Measurement...of tests?

It is unclear that how are "nanocomposites". Explain clear what it define and confirm nano in your data? 

Some words/constructions are technical inappropriate e.g. Aids(aids), S/A, table 1. Formation of ?....Figure 1. Schematic structure diagram?

Author contributions?: It is just one! Make a correction.

Author Response

Dear Editor,

We thank the reviewers for their comments and kind suggestions. We are also glad to send this clarification report to the editor and reviewers. Our manuscript has been explicitly revised in accordance with the reviewers’ comments and suggestions, and the modifications are marked with red color in the revised manuscript.

We believe that these modifications make our manuscript more appropriate for publication in Applied Sciences. We also anticipate that the novel proposed approach and results presented in this manuscript will be of significant interest to the general readership of the journal.

The response letter for addressing the reviewers’ comments consists of 4 pages.

Thank you and best regards,

Yours sincerely

Prof. Sung Ho Song

Reviewer 2

Comments: The manuscript present an influence of some chemical compounds in the obtaining of rubber composites to improve dynamic mechanical properties. Despite of interesting data, figures and discussion there are and some gaps from my point of view.

Abstract:10-13...zinc? inappropriate...that in manuscript is not so developed about it...

Response for question

Thank you for your comments. Generally, in tire industry, as a lubricant and dispersing agent, processing aids containing zinc ions have been employed as additives. In this manuscript, as zinc is a heavy metal, alternative eco-friendly processing aids are required to satisfy worldwide environmental concerns. Therefore, we studied non-toxic, degradable, and renewable processing aids to improve the mechanical properties of the rubber composites. Processing aids containing Zn ions are mentioned simply to highlight the need for green processing aids.

Introduction: at least form 70 it is necessary to rewrite the text. You have already here and conclusions (77-80) 

Response for question

Introduction Part:

In this study, we investigated the effects of new silica-based processing aids, utilizing terpene phenol resin (TPR), which have hydrophilic and hydrophobic groups at opposite ends of the molecule.

2.4. Measurement...of tests?

It is unclear that how are "nanocomposites". Explain clear what it define and confirm nano in your data? 

Response for question

Except for the reference, there is no mention of nanocomposite in this paper.

Some words/constructions are technical inappropriate e.g. Aids(aids), S/A, table 1. Formationof ?....Figure 1. Schematic structure diagram?

Response for question

As the reviewer’s comment, we correct the words in the manuscript.

In Table: Stearic acid => S/A

Formation => Formulation

Table 1. Formulation of rubber composites with varying content fraction of different resins.

Schematic structure diagram => Molecular structure diagram

Figure 1. Molecular structure diagram of a) Hydrocarbon resin (HCR), b) Alpha Methyl Styrene Resin (AMR), c) Alkyl Phenol Resin (APR), d) Terpene Phenol Resin (TPR).

Author contributions?: It is just one! Make a correction.

Response for question

The authors declare no conflict of interest.

Round 2

Reviewer 1 Report

The authors improved easy points, including English expressions, without explaining their revision to the reviewer. The most concerning issue was “References.” I pointed out several inappropriate self-citations in the previous reports, but the authors never improved. The most profound subsection, “2.2 Preparation of Rubber Composites,” keeps the almost original style.  A typing error in eq. 5 remains as it was, and the wrong usage of “scorch time” in L139 was unchanged. In addition to them, the figures also have not been changed at all, although only the tables were rewritten. However, Table 4 still shows an unacceptable condition. Did the authors check the manuscript after their revision procedure? The TEM photo was added, which is preferable, but the figure number seems to be inappropriate. The used TEM system’s information is wrong. JEOL is a Japanese company, not a Korean. As a whole of the revised manuscript, I get the impression that it still has several serious problems.

Author Response

Dear Editor,

We thank the reviewers for their comments and kind suggestions. We are also glad to send this clarification report to the editor and reviewers. Our manuscript has been explicitly revised in accordance with the reviewers’ comments and suggestions, and the modifications are marked with red color in the revised manuscript.

We believe that these modifications make our manuscript more appropriate for publication in Applied Sciences. We also anticipate that the novel proposed approach and results presented in this manuscript will be of significant interest to the general readership of the journal.

The response letter for addressing the reviewers’ comments consists of 7 pages.

Thank you and best regards,

Yours sincerely

Prof. Sung Ho Song

Reviewer 1

Comments: The authors improved easy points, including English expressions, without explaining their revision to the reviewer. The most concerning issue was “References.” I pointed out several inappropriate self-citations in the previous reports, but the authors never improved.

Response for question 1

All self-citation papers were changed to other papers.

  1. Essabir, H.; Raji, M.; Mokhtar, E.; Rodrigue, D.; Gouhfid, R.; Qaiss, A. e. Morphological, thermal, mechanical, electrical and magnetic properties of ABS/PA6/SBR blends with Fe3O4 nanoparticles. J. Mater. Sci. Sci. 2017, 28, 17120-17130.
  2. Yang, S.; Liang, P.; Peng, X.; Zhou, Y.; Hua, K.; Wu, W.; Cai, Z. Improvement in mechanical properties of SBR/Fly ash composites by in-situ grafting-neutralization reaction. Macromol. Chem. Eng. J. 2018, 354, 849-855.
  3. Liu, Z.; Zhang, Y. Enhanced mechanical and thermal properties of SBR composites by introducing graphene oxide nanosheets decorated with silica particles. Compos. Part A Appl. Sci. Manuf. 2017, 102, 236–2428.
  4. Guo, H..; Jerrams, S.; Xu, Z.; Zhou, Y.; Jiang, L.; Zhang, L.; Liu, L.; Wen, S. Enhanced fatigue and durability of carbon black/natural rubber composites reinforced with graphene oxide and carbon nanotubes. Eng. Fract. Mech. 2020, 223, 106764.

  1. 2. The most profound subsection, “2.2 Preparation of Rubber Composites,” keeps the almost original style. 

Response for question 2

I checked the table in the manuscript again, but the contents related to the table were well mentioned in the manuscript. Also, according to the MDPI format, the table cannot be attached as a picture file, making it difficult to modify. Therefore, picture files are attached for reviewer’s understanding.

Table 1. Formulation of rubber composites with varying content fraction of different resins.

                                                                                             phr

Furthermore, the formulation method applied in this paper has already been verified and published in many papers. Related references are attached.

Polymer Engineering & Science 58 (10), 1843-1848

Applied Science polymer Science 134(40), 45376

RSC advances 7 (72), 45595-45600

Composite Interfaces 25 (10), 919-927

Journal of Industrial and Engineering Chemistry 66, 78-85

International Journal of Polymer Science 2019

Macromolecular Research 27 (9), 850-856

  1. A typing error in eq. 5 remains as it was, and the wrong usage of “scorch time” in L139 was unchanged.

Response for question 3

Equation 5 is a general equation, it is not wrong, and other papers and related parts are attached.

Reference: ACS Sustainable Chem. Eng. 2018, 6, 325−336

Related parts:

Reference: Macromol. Res., 27(11), 1136-1143 (2019)

Related parts:

Reference: Materials Science and Engineering 426 (2018) 012006

Related parts:

We have modified the scorch time part.

L139: on cure time (t40 and t90),

L142: difference in the curing time

  1. In addition to them, the figures also have not been changed at all, although only the tables were rewritten.

Response for question 4

According to the MDPI format, the table cannot be attached as a picture file, making it difficult to modify. Therefore, the form of a picture file to help reviewers understand. Please refer to the MDPI form. Therefore, picture files are attached for reviewer’s understanding.

Table 2. Swelling and cross-linking density characteristics of rubber composites with varying content fraction of different resins.

Table 3. Parameter values of rubber composites with varying content fraction of different resins.

Table 4. Fatigue properties of rubber composites with varying content fraction of different resins.

  1. The TEM photo was added, which is preferable, but the figure number seems to be inappropriate.

Response for question 5

Since there is no part showing microstructure in the paper, it is attached as supporting information rather than main figure. In addition, MDPI does not matter if the supporting information is added.

  1. The used TEM system’s information is wrong. JEOL is a Japanese company, not a Korean. As a whole of the revised manuscript, I get the impression that it still has several serious problems.

Response for question 6

Transmission electron microscopy (TEM, JEOL JSM-2100, Japan) was used to observe the morphology of the fractured surface of rubber composites to confirm the dispersibility of silica.